# Factors Limiting Shelf Life of a Tomato–Oil Homogenate (Salmorejo) Pasteurised via Conventional and Radiofrequency Continuous Heating and Packed in Polyethylene Bottles

**DOI:** 10.3390/foods12203882

**Published:** 2023-10-23

**Authors:** Marina Kravets, Andrés Abea, Maria Dolors Guàrdia, Israel Muñoz, Sancho Bañón

**Affiliations:** 1Department of Food Technology and Science and Nutrition, Veterinary Faculty, Regional Campus of International Excellence “Campus Mare Nostrum”, University of Murcia, 30100 Murcia, Spain; marina.kravets@um.es; 2Food Technology Program, Institut de Recerca i Tecnologia Agroalimentàries IRTA, Finca Camps i Armet, Monells, 17121 Girona, Spain; andres.abea@irta.cat (A.A.); dolors.guardia@irta.cat (M.D.G.); israel.munoz@irta.cat (I.M.)

**Keywords:** fresh-like, tomato, deterioration, stability, pasteurisation, dielectric heating, refrigeration

## Abstract

Salmorejo is a tomato–oil cold puree commercialized as a “fresh-like” product requiring mild pasteurisation and chill storage to reach a suitable shelf lifetime. The objective of this study was to study the factors which limit the shelf life of salmorejo pasteurised via conventional or radiofrequency continuous heating, packed in high-density polyethylene bottles, and kept at refrigeration. The pasteurised–chilled salmorejo reached a long shelf life (4 months) compared to that of pasteurised tomato juices or purees. Mesophilic and pathogenic bacteria were easily inhibited in this acidic product. Salmorejo mainly showed oxidative and subsequent sensory changes. Initial enzyme oxidation was associated with some adverse effects (loss of vitamin C and lipid oxidation) at the first month, although there were no sensory implications. Salmorejo remained stable at the physicochemical and sensory levels for the following 3 months, though colour and viscosity changes could be measured with instruments. Between the fourth and fifth month, salmorejo showed clear signs of deterioration, including changes in appearance (slight browning and loss of smooth surface), odour/flavour (loss of freshness and homogenisation), and consistency (thinning trend). The shelf life of salmorejo is limited by long-term oxidative deterioration and their sensory implications.

## 1. Introduction

Salmorejo is a Spanish traditional cold purée prepared with tomato, olive oil, breadcrumbs and other thickening ingredients, vinegar, garlic, and salt. The production of salmorejo and gazpacho (a similar product made with a different recipe in the same manufacturing lines) is growing in Spain (about 96 million litres in 2021), and it is now exported to other European countries [1]. Some international companies produce “fresh-like” salmorejo pasteurised with HTST (High-Temperature Short-Time) treatments. Since the 1990s, salmorejo has been packed in Tetra Brik containers though the use of plastic bottles, which is now increasing. Among the available plastic materials, high-density polyethylene (HDPE) is being widely used in juice, milk, and water bottles because it is a suitable material for long-term storage as it is one of the most stable and inert plastics [2]. The temperature of freshly pasteurised product must be decreased before aseptic filling since HDPE bottles can be deformed under the influence of heat [3]. Packed salmorejo is commercialized in refrigerated display cabinets for up to several months [4]. Salmorejo can reach this long shelf lifetime as it is traditionally prepared with vinegar and bread that act as acidifying and thickening agents, respectively, which contribute to its stabilization. Using mild pasteurisation treatments can be sufficient in ensuring the microbiological quality of salmorejo or gazpacho as spoilage and pathogen microorganisms cannot easily proliferate in tomato products of a low pH [5,6,7]. However, as with other tomato products, salmorejo tends toward clarification, browning, and flavour alteration enhanced by enzymes responsible for tomato ripening, including pectinmethylesterase (PME), polygalacturonase (PG), polyphenoloxidase (PPO), or peroxidase (POD) [8]. Therefore, pasteurisation treatments which can inactivate these enzymes are required to obtain salmorejo with an extended shelf life.

Pasteurisation conditions reported for “fresh-like” gazpacho range from 80 to 90 °C for 1 min, resulting in products with a shelf life of 30–60 days in refrigeration [9,10]. High-pressure processing has also been tested to stabilize gazpacho and salmorejo with better results in terms of microbiological rather than enzymatic inactivation [6,9,10]. To date, the physical, chemical, and sensory changes determining the shelf life of this type of pasteurised tomato-based homogenates have not been well established. Salmorejo differs from tomato juices or purees in that it is an oil–tomato thickened emulsion with a typical orange colour and a complex odour and flavour where vinegar and garlic also contribute to freshness, a key sensory aspect for the acceptability of the product [11]. Therefore, salmorejo cannot be elaborated with blanched tomato and must be carefully pasteurised as its sensory attributes are easily altered. Heating can bring some sensory changes, including the formation of cooked fruit flavour related to sulphur volatiles and others [12,13], discolouration due to the degradation of lycopene and other pigments [14], or an increase in consistency due to pectin transformations [15]. These changes lead to products acquiring the sensory characteristics of canned heated tomato, undesirable for minimally pasteurised products such as salmorejo, gazpacho, or grated fresh tomato.

Dielectric heating by radiofrequency (RF) has recently been tested to improve the pasteurisation methods for these types of tomato–oil emulsions. RF technology could replace conventional heat exchangers in the food industry as dielectric heating has better thermal penetration and could be less energy-intensive [16,17]. A preliminary trial conducted by Kravets et al. (2023) [18] confirmed that “fresh-like” salmorejo can be pasteurised using either conventional heat exchangers or dielectric equipment. Applying 80 °C for around 10 s in a continuous line with a preheating stage provides an adequate decimal reduction in mesophilic bacteria and foodborne pathogens such as *L. monocytogenes*. These HTST conditions also inactivated PME but did not completely inactivate PPO and POD and did not affect PG enzyme, while increasing the pasteurisation temperature up to 100 °C did not improve the results of enzyme inactivation, and there were some signs of overheating. This is of great relevance since “fresh-like” salmorejo should retain sensory traits like those of a fresh raw product for several months The research hypothesis was that HTST salmorejo mainly deteriorates due to oxidation phenomena enhanced, to a greater or lesser degree, by the tomato enzymes that remain active. The time that these enzymes can remain active in the pasteurised–chilled salmorejo has not yet been determined. Salmorejo presents some properties that might contribute to its oxidative stability: (i) tomato is rich in lycopene and other phenolic antioxidants; (ii) it is made with extra virgin olive oil rich in α-tocopherol and other lipid antioxidants; and (iii) this emulsion is thickened with breadcrumbs or other ingredients, which may delay clarification signs.

The objective of this study was to study the factors which limit the shelf life of pasteurised–chilled salmorejo. Different microbiological, enzyme, physicochemical, and sensory aspects were studied in a salmorejo alternatively pasteurised via conventional (CH) or radiofrequency (RF) continuous heating, packed into HDPE bottles, and kept under refrigeration for up to five months.

## 2. Materials and Methods

### 2.1. Salmorejo Manufacturing Process

A salmorejo with fibre was formulated with a partial substitution of breadcrumbs for inulin (Orafti^®^ high-polymerizate HP inulin, Beneo, Barcelona, Spain). Salmorejo ingredients (g·100 g^−1^) were vine tomatoes (87.6), extra virgin olive oil (Var. Hojiblanca) (5.0), breadcrumbs (3.0), inulin (2.0), wine vinegar (1.5), garlic (0.7), and salt (0.2). Briefly, tomatoes were ground in a cutter and sieved (3 mm diameter). Tomato purée was then mixed with olive oil, breadcrumbs, salt, vinegar, inulin, and garlic by stirring in a tank for 30 min. Raw salmorejo was homogenised in an MZ-100 colloid mill (FrymaKoruma AG, Rheinfelden, Switzerland) and preheated at 50 °C for 15 s in a tubular heat exchanger. Before pasteurisation, preheated product passed through a high-pressure homogeniser (HA31002, Bertoli SRL, Parma, Italy), working at 250 bar. The thermal treatment was targeted to inactivate *Escherichia coli* O157:H7, *Salmonella enterica*, and *Listeria monocytogenes* to ensure at least a 5-log reduction in vegetative pathogens of concern. Applying 80 °C for around 10 s in a continuous line with a preheating stage provides an adequate decimal reduction in mesophilic bacteria and foodborne pathogens. This temperature was determined using the Z-values corresponding to an unacidified tomato purée at 4.5 pH. A minimum cumulative total lethality equivalent to *p* = 0.51 min at 71.1 °C was needed for a cocktail of pathogens [19]. For more details, see Kravets et al. (2023) [18]. Salmorejo was alternately pasteurised at 80 °C using continuous-flow (200 L·h^−1^ of product) in a conventional heat exchanger or in RF equipment. For CH, it was heated in a 2-stage tubular heat exchanger built by Inoxpa (Banyoles, Barcelona, Spain) with a holding time of 10.9 s. For RF, it was heated using a 45 kW EVO RF Cartigiliano S.p.A (Cartigliano, Vicenza, Italy) working at 27.12 MHz with a holding time of 9.08 s and a maximum temperature increment of 50 °C. After processing, the product was cooled to 25 °C using a tubular heat exchanger and aseptically packaged in 250 mL translucent HDPE bottles with white polypropylene closures (Cat. No.342089; Nalguene narrow-mouth bottles; Thermo Fisher, Barcelona, Spain). According to the supplier, this sterilised bottle has excellent thermal (−100 to +120 °C) and chemical resistance. For the shelf life study, bottled samples were kept at 4 °C in darkness for up to 5 months at one-month intervals. All samples were analysed in triplicate.

### 2.2. Microbiological Analyses

Total mesophilic aerobic (TMA) microorganisms were enumerated on Plate Count Agar (PCA, Scharlab, Barcelona, Spain) after incubation at 30 °C for 48–72 h (International Organization for Standardization) [20]. Mesophilic spores were enumerated after heat treatment at 80 °C for 10 min and plating on PCA and incubated at 30 °C for 24–48 h [21]. The detection limit was 1 Log·CFU g^−1^. The log reduction in TMA and spores was quantified as the difference between count (Log·CFU g^−1^) samples before and after the treatments (pasteurisation + storage time).

### 2.3. Enzyme Relative Activities (%RE)

%RE were determined with respect to the unpasteurised product. PME activity was determined according to Arjmandi et al. (2017) [22]. A total of 2.5 mL salmorejo was mixed with 10 mL 0.2 M NaCl solution and filtered with cotton to remove the fraction not soluble in water. Then, 2.5 mL filtrate was mixed with 15 mL of a 1% (*w*/*v*) citrus pectin solution containing 0.2 M NaCl solution. The pH was adjusted to 7.0 using some drops of 0.2 M NaOH solution. The sample was titrated with 0.01 M NaOH solution for 10 min using a GLP21 pH meter (Crison, Barcelona, Spain). PME activity was calculated as PME units mL^−1^. PG activity was determined according to Fachin et al. (2003) [23]. In total, 3 mL salmorejo was mixed with 3 mL acidified water (37% HCl; pH 3) at 4 °C in a RX3 vortex mixer (Velp Scientifica™, Usmate Velate MB, Italy). The mix was kept under agitation in a Rotavit (JP Selecta™, Barcelona, Spain) at 170 u·min^−1^ and 4 °C for 20 min in darkness, and was then centrifuged in a Digicen 21 R centrifuge (Orto Alresa, Madrid, Spain) at 4554× *g* and 4 °C for 10 min. Sample precipitate was reconstituted until 6 mL with a 1.2 M NaCl solution, shaken (at 170 u·min^−1^ and 4 °C for 1 h), and centrifuged (at 4554× *g* and 4 °C for 10 min) again. The supernatant was filtered through Whatman No. 1 paper to obtain enzyme extract. Then, 100 µL enzyme extract was mixed with 300 µL 0.2% (*w*:*v*) polygalacturonic acid (Sigma, St. Louis, MO, United States) and incubated in a water bath at 35 °C and in darkness for 10 min. To stop the reaction, the sample was first mixed with 2000 µL 0.1 M borate buffer at pH 9 (composed of Na_2_B_4_O_7_ 10 H_2_O and H_3_BO_3_ at 1.47/0.38 (*w*:*w*) and 400 µL 1% (*w*:*v*) cyanoacetamide as substrate, and then incubated in a water bath at 100 °C for 10 min. The sample was cooled at room temperature, transferred to 3.5 mL quartz cuvettes, and measured at 276 nm in a UV-Vis Genesys™ 180 spectrophotometer (Thermo Fisher Scientific, Madison, WI, USA).

POD activity was determined according to Vervoort et al. (2012) [24]. A total of 3 mL salmorejo was mixed with 6 mL 0.2 M sodium phosphate buffer (pH 6.5) and centrifuged at 4554× *g* and 4 °C for 15 min. The sample was filtered with Whatman No. 1 paper to obtain enzyme extract. Then, 0.25 mL enzyme extract was mixed with 1.1 mL 0.2 M sodium phosphate buffer (pH 6.5), 0.5 mL O-Phenylenediamine solution (10 g L^−1^ in 0.2 M sodium phosphate buffer pH 6.5) as substrate, and 0.25 mL hydrogen peroxide solution (15 g L^−1^ in 0.2 M sodium phosphate buffer pH 6.5) as oxidising reagent. The sample was kept at 25 °C and in darkness for 20 min, and absorbance was then measured at 485 nm. The enzyme extract was replaced by distilled water to obtain the blank. PPO activity was determined according to Marszałek et al. (2015) [25]. In total, 3 g salmorejo was mixed with 6 mL 0.2 M sodium phosphate buffer (pH 7.0) containing 10 g L^−1^ insoluble Polyvinylpyrrolidone and 5 g L^−1^ Triton X-100. The mix was centrifuged at 4554× *g* and 4 °C for 30 min and was filtered with Whatman No. 1 paper to obtain enzyme extract. A total of 75 µL enzyme extract was mixed with 1 mL 0.21 M catechol solution (0.05 M sodium phosphate buffer: pH 6.5) as substrate and 2 mL 0.2 M sodium phosphate buffer and measured at 420 nm and 25 °C. PPO activity was calculated using a linear regression obtained from the absorbance measured each min during 30 min.

### 2.4. Physicochemical Assessment

Ascorbic acid (mg 100 mL^−1^) was determined by HPLC-DAD [26]. In total, 3 g salmorejo was completed with HPO_3_ 4.5% *(w*/*v*) water solution until it reached 10 mL in a volumetric flask. The mixture was stirred for 1 min and then centrifuged at 4053× *g* and 4 °C for 10 min. The supernatant was filtered with Whatman No. 1 filter to obtain sample extract. A total of 1.5 mL sample extract was filtered with a 0.45 µm Millipore SAS filter and injected into an Infinity II HPLC/DAD-1260 Series system (Agilent Technologies, Santa Clara, CA, USA). A Brisa LC2C18 column (25 × 0.46 cm, with 5 µm pore size) was used (Teknokroma, Barcelona, Spain). Operating conditions were (i) flow rate: 0.8 mL min^−1^; (ii) mobile phase: acidified water (2.5% sulfuric acid *v*/*v*); (iii) temperature: 35 °C; (iv) analysis time duration: 20 min; and (v) detection wavelength: 245 nm. A total of 0.025 g L-(+)-ascorbic acid (Panreac, Castellar del Vallès, Spain) was mixed with a HPO_3_ 4.5% (*w*/*v*) water solution until it reached 50 mL in volumetric flask to prepare standard solution. A calibration line (y = 39024x − 15.659; R^2^ = 0.9999) of this standard solution at concentrations from 50 to 1200 mg ascorbic acid per mL was used for quantification.

Thiobarbituric acid reactive substances (TBARSs) were determined according to Botsoglou et al. (1994) [27]. In total, 4 g salmorejo was mixed with 8 mL trichloroacetic acid (TCA) water solution (5% *w*/*w*) and mL butylated hydroxytoluene (BHT) hexane solution (0.8 *w*/*v*) as an antioxidant. The mix was centrifuged at 1328× *g* and 4 °C for 10 min. Approximately 8 mL supernatant was given to 10 mL with a TCA water solution (5% *w*/*v*) and centrifuged again (at 1328× *g* and 4 °C for 10 min) to improve sample turbidity. Then, 2.5 mL supernatant was mixed with 1.5 mL thiobarbituric acid (TBA) water solution (5% *w*/*v*) and incubated in a water bath at 70 °C for 30 min. Sample absorbance was measured at 532 nm and 25 °C. Sample was replaced by TBA water solution to obtain the blank. Malondialdehyde (MDA) standard calibration curve was prepared with 1,1,3,3-tetraethoxypropane Sigma-Aldrich) 0.1 M HCl water solution (*v*/*v*) at concentrations ranging 0.1–10 μM (y = 1171x + 0.0038; R^2^ = 1). Results were expressed as mg MDA kg^−1^.

The pH was measured with a GLP21 pHmeter (Crison, Barcelona, Spain) equipped with a combined electrode Cat. No. 52-21 (Ingold Electrodes, Wilmington, NC, USA). CIELab colour was measured using a CR-200/08 Chroma Meter II (Minolta Ltd., Milton Keynes, UK) with a D65 illuminant, 2° observer angle and 50 mm aperture size. Results were expressed as CIELab values: lightness (L*), redness (a*), yellowness (b*), hue angle (H* = tan^−1^ b*/a*), and ΔE* = [(L*_month n_ − L*_month n+1_)^2^ + (a*_month n_ − a*_month n+1_)^2^ + (b*_month n_ − b*_month n+1_)^2^]^1/2^. Dynamic viscosity was measured using two different procedures. Viscosity (V1) (Pa s) was measured in samples sheared for 1 min with a standard cylindrical vessel 125 × 28 mm (head space, 20 mm) using a 6 L/plus rotational Viscotester (Haake, Thermo Scientific, Karlsruhe, Germany). The measuring conditions for V1 were 80 mL sample; L3 probe; 100 rpm angular speed; 4 °C temperature. Viscosity (V2) (mPa s), including initial shear stress, was measured at 20 °C using an MV1 cylindrical probe in a Haake 550 rotational viscometer (Thermo Scientific, Waltham, MA, USA). Flow curves were obtained from stepped shear stress ramp (steady state approximation: 15 s per point). Ranges of shear stresses, in lineal distribution, were used to obtain shear rates between 450 and 1800 s^−1^. Flow curves data were fitted to the Bingham model (τ = τ0B + γμpl), where τ is the shear stress [Pa], γ˙ the shear rate [s^−1^], τ0B the Bingham yield stress [Pa], and μpl the plastic viscosity [Pa s]. Texture was analysed using a QTS-25 Analyser (Brookfield, Harlow, Essex, England). Selected parameters were hardness (N), the maximum force required to compress the material, and deformation energy (mJ), the energy required to deform the sample. Operating conditions were 5 °C; TA3/100 flat cylindrical probe (20 mm in diameter); trigger point, 0.05 N; compression objective, 5 mm; crosshead speed, 0.1 mm s^−1^; and charge cell, 10 kg.

### 2.5. Sensory Analysis

A quantitative descriptive analysis (QDA) of the salmorejo samples was performed in 10 sessions/replica. The generation of the descriptors was carried out by open discussion and consensus in two previous sessions using both untreated and treated samples (CH and RF). Six selected and trained panellists (International Organization for Standardization, refs. [28,29]) performed the sensory analysis on 50 mL of salmorejo sample using a complete block design. A non-structured scoring scale was used, where 0 and 10 meant absence and high intensity of the attribute, respectively. The selection of descriptors included in the final sensory profile was based on those that help to discriminate among samples. The sensory descriptors used were (i) orange colour; (ii) smooth surface (homogeneous and soft surface); (iii) overall odour/flavour; (iv) fresh odour/flavour (which corresponds to a freshly made product); (v) tomato odour; (vi) vinegar odour/flavour; (vii) garlic odour; (viii) acidity (basic taste sensation elicited by citric acid); (ix) mouth-feel (degree to which juice is thick, coats the mouth, and difficulty to swallow); and (x) viscosity with spoon (difficulty with which sample flows using a teaspoon). Samples were coded with three-digit random numbers and presented to assessors balancing the first order and the carry over effect. The average score of each sample and session was recorded and used in data analysis.

### 2.6. Statistical Analysis

A randomised design was performed with the storage time (0–5 months) and heating method (CH or RF) as treatments. The effects of the treatments on the dependent variables were determined using a two-way ANOVA (Repeated measures design; Tukey range test; *p* < 0.05). Sample size was n = 108 (3 bottled samples of 250 mL × 2 heating methods × 6 retailing times × 3 manufacturing batches). Data were analysed using the Statistix 8.0 software for Windows (Analytical Software, Tallahassee, FL, USA).

## 3. Results

Pasteurisation treatments (either CH or RF) largely ensured the microbiological quality of salmorejo kept at 4 °C for 5 months (Figure 1). Initial TMA counts were similar in the CH (4.15 Log·CFU g^−1^) and RF (3.90 Log·CFU g^−1^) untreated salmorejo. Log reductions in TMA counts reached through pasteurisation were higher for CH (over 1 Log·CFU g^−1^) than for RF (over 0.5 Log·CFU g^−1^) salmorejo and were not affected by storage time. Mesophilic spore counts (Log·CFU g^−1^) were similar in CH (2.51 Log·CFU g^−1^) and RF (2.98 Log·CFU g^−1^) untreated salmorejo. Pasteurisation allowed a higher Log reduction in mesophilic spores in CH (0.35–1.00 Log·CFU g^−1^) than in RF (0.05–0.75 Log·CFU g^−1^) salmorejo with some oscillations over time, although with a similar trend. Chill storage clearly inhibited the proliferation of mesophilic microorganisms in pasteurised salmorejo. In contrast, pasteurised salmorejo (either CH or RF) showed some enzyme activities during its shelf life (Figure 2). PG was the most heat resistant enzyme, followed markedly by PPO, POD, and PME enzymes. Pasteurisation completely inhibited PME activity and slightly enhanced PG activity (>100% RA) throughout storage. POD activity remained over 11–12% RA at the beginning of storage, decreased by half at the first month, and then remained stable over 3% during the rest of storage. PPO initial activity was lower in the CH (20.5% RA) than in the RF (16.0% RA) salmorejo; both values decreased to 6% RA at the first month and then decreased to lower values (1.2–0.0% RA). Therefore, salmorejo was more prone to enzyme spoilage at the beginning of chill storage.

Physicochemical changes during shelf life were similar in CH and RF salmorejo, though with some exceptions (Table 1). The pH value remained stable over 3.8–4.0 for four months and then increased to 4.5 at the fifth month. Ascorbic acid levels (mg 100 mL^−1^), used as a general oxidation index, were similar at the first month in the CH (19.6 mg 100 mL^−1^) and RF salmorejo (21.5 mg 100 mL^−1^), and then practically disappeared (<1.3 mg 100 mL^−1^). TBARS values, used as the lipid oxidation index, increased at the first month (>0.57 mg MDA kg^−1^) and then tended to decrease (up to 0.45 mg MDA kg^−1^) during the rest of storage. Both ascorbic acid and TBARS values confirmed that pasteurised salmorejo oxidized to a greater extent at the beginning of chill storage, coinciding with the period of higher activity of oxidase enzymes. The chromatic changes over time were modest. L* values remained quite stable over 59 CIE units during storage, with no clear signs of darkening. In contrast, the H* angle increased at the second month (from 52 to 66 CIE units), indicating the development of brown tones, and then remained stable. This succeeded because b* values increased while a* values decreased. ΔE values (Figure 3) confirmed that major chromatic changes occurred between the second and third month of storage. Apparent viscosity measured, including the initial stress (V1), tended to decrease at the second month (from 22–25 to 17–18 mPa s) and then remained stable, while, similarly, apparent viscosity measured in sheared samples for 1 min (V2) tended to decrease during storage. The highest and lowest values of V2 were recorded at the beginning (over 0.81 Pa s) and the end of storage (over 0.55 Pa s). The effects of chill storage on instrumental texture data (deformation energy and hardness) were less clear, and there was no clear thickening tendency.

Most sensory changes detected in salmorejo (CH or RF) occurred at the end of storage (Table 2). The appearance of salmorejo is characterized by an orange colour and smooth surface due to tomato–oil emulsion. The intensity of the orange colour was stable during storage (>6.0 in a ten-point scale) but slightly decreased at month 5 (<5.5) and was associated with a lower intensity of the smooth surface. Overall odour and flavour are generated by the homogenisation of its ingredients (tomato, vinegar, garlic, etc.), which could also be perceived individually. Both attributes were not affected by storage time. Changes in the general intensity of the odour (4.7–5.9) and flavour (5.2–6.0) over time were not relevant. Fresh odour and flavour intensity were weak (over 4.6) and stable during most of the storage period, decreasing at the fifth month (<3.4). This loss of freshness was often associated with a lesser intensity of tomato, vinegar, garlic, and acidity odour/flavour scores. Mouth feeling consistency, like a puree, also scored low (<4.3) from the beginning of storage, and a certain loss of consistency (<2.8) was detected at month 5, coinciding with the general deterioration of the product. According to the sensory criteria, the shelf life of pasteurised–chilled salmorejo was established in 4 months.

## 4. Discussion

Tomato–oil homogenates such as salmorejo or gazpacho are considered to be products of low microbiological risk. As seen, mesophilic microorganisms were inhibited throughout chill storage. Oscillations observed in spore counts could be explained by the biological–microbial variability itself. In ‘naturally’ contaminated matrices (not inoculated) such as salmorejo, the type of microorganisms can vary and be heterogeneously distributed. The magnitude of an increase in bacterial count may be statistically significant but not be biologically relevant. It is generally considered that the uncertainty (extended uncertainty) associated with microbial load determination procedures can range between 0.5 and 1 Log depending on the technique requiring colony confirmation [30]. In the specific case of *L. monocytogenes*, the criterion of 0.5 Log is considered appropriate in the European Union, equivalent to twice the estimated standard deviation (0.25) associated with the plate colony count [31]. Similarly, according to the criteria used, growth inhibition would occur when the increase in the bacterial count was less than 0.5 or 1 Log. Therefore, small microbiological differences should not have any impact on the safety and/or quality of the product. In salmorejo, different synergistic factors, including natural acidity, heating, and refrigerated storage, likely contributed to maintaining microbiological quality. Salmorejo was elaborated with fresh tomato, acidifying (wine vinegar), olive oil, salt, and thickening polysaccharides (bread starch and chicory inulin) being a product of low pH (pH = 4) and with high water activity (0.95). HTST pasteurisation at 80 °C (either CH or RF) reduced the initial loads of mesophilic microorganisms (by 0.5–1.1 Log) and their spores (by 0.1–0.5 Log). Log reductions in TMA counts were quite similar at the beginning and end of storage, meaning that chilled product was microbiologically stable.

Salmorejo is an acidic product which, from a microbiological standpoint, could even be manufactured without pasteurising. The pH values reported for commercial and homemade salmorejo range from 3.9 to 4.4, while their water activity is near 1 [32]. Both values depend on the quantities of vinegar and oil used in the recipe. Toledo Del Árbol et al. (2015) [6] reported that mesophilic bacteria counts decreased from 4.3 to 2.1 Log·CFU g^−1^ in untreated salmorejo (pH = 4.0) kept at 4 °C for 30 days, meaning that these bacteria can be naturally inhibited in this product. In contrast, Quintín (2015) [5] found that mesophilic bacteria counts increased from 1.3 to 3.5 Log·CFU g^−1^ in untreated gazpacho (pH = 4.2) kept at 5 °C for 60 days; however, mould and yeast counts were above 1 Log·CFU g^−1^, and natural bacteria counted as hygiene indexes (*L. monocytogenes*, *E coli*, or *Salmonella* spp.) did not proliferate. Other authors agree that a restricted range of moulds, yeasts, and lactic acid bacteria can only proliferate in tomato products of low pH [7,19]. Specific information on the microbial shelf life of pasteurised–chilled salmorejo is scarce. Toledo Del Árbol et al. (2015) [6] confirmed that foodborne pathogens bacteria such as *L. monocytogenes*, *S. enterica*, and *E. coli* O157 cannot proliferate when inoculated in salmorejo kept at 4 °C for up to 30 days, being easily inhibited by pasteurisation by high-pressure processing (600 MPa); this pressurization treatment also decreased the counts of background microbiota (from 4.3 to 1.8 Log·CFU g^−1^) from salmorejo not inoculated with pathogens, which, as in the present study, remained stable during further shelf life. Similarly, Daoudi (2004) [9] did not find mesophilic or lactic bacteria, mould, or yeasts in gazpacho pasteurised using heat plus high pressure (400–500 MPa at 45–50 °C for 15–30 min) and kept refrigerated for 30 days. According to the above information, microbiological events do not seem to play a relevant role in the shelf life of “fresh-like” salmorejo.

Cloud clarification related to pectolytic enzymes is often considered a deterioration sign in many stored fruit juices. PME and PG are involved in the breakdown of pectins [33]; PME catalyses the removal of methyl groups from the polygalacturonic acid chain, increasing the number of free carboxyl groups that can then bind cations and cross-link pectin chains. The latter can aggregate and settle, leading to cloud clarification. PG acts only on segments of the pectin chain that have been demethylated by PME, contributing to a reduction in juice viscosity. As seen, PME was completely inactivated upon HTST conditions, while PG remained unaltered. The thermostable fraction of PG would be responsible for its residual activity in tomato products [23]. Several studies on tomato agree that PME is more thermolabile than PG enzyme, which requires prolonged thermal treatments to be inactivated [23,33,34,35,36]. In the present study, PG activity was not affected by refrigerated storage, though, in theory, complete inactivation reached for the PME might have restricted the PG pectolytic activity, attenuating the clarification of salmorejo. In other studies, these enzymes followed a different pattern. Arjmandi et al. (2016) [35] reported respective decreases in the remaining activities of PME (from 28% to 10% RA) and PG (from 70% to 45% RA) in a pasteurised (90 °C for 35 s) smoothie with tomato stored at 5 °C for 45 days; however, in another study on pressurized-chilled gazpacho, PME activities maintained stable over 20% RA for 30 days [9].

Unlike stored fruit juices which can lose their natural turbidity, salmorejo is a tomato–oil thickened emulsion whose turbidity cannot be measured with a nephelometer (>4000 NTU), so a clarification trend was objectively assessed through flow behaviour (apparent viscosity) and texture parameters (deformation energy and hardness). This information was contrasted with the sensory assessment of consistency. As observed, a decrease in apparent viscosity was recorded at the beginning of storage, while texture changes over time were barely relevant. Sensory signs of clarification trend (lesser consistency assessed with mouth and spoon) were perceived at the end of storage. From a sensory viewpoint, early clarification does not appear to limit the shelf life of salmorejo, a thickened product, though the product turned less consistent as storage time and oxidation advanced, which contributed, together with other sensory alterations, to overall sensory deterioration. In gazpacho, a product less consistent than salmorejo, small changes in apparent viscosity and flow behaviour indexes during shelf life (30 days) could not be detected in the sensory assessment [9].

Oxidase enzymes also contribute to the deterioration of tomato products during shelf life. PPO catalyses the oxidation of the functional OH group attached to the carbon atom of the benzene ring of monohydroxy phenols to O-dihydroxy phenols and the dehydrogenation of O-dihydroxy phenols to O-quinones, which are highly reactive compounds that polymerize rapidly to form brown pigments that cause enzymatic browning [37]. POD belongs to a group of oxidases that use H_2_O_2_ as a catalyst for the oxidation of phenolic compounds and other substrates related to undesirable changes in flavour, texture, and colour [37]. Tomato PPO exhibits maximum activities at pH 4.8 [38], while tomato POD exhibits its maximum activity at pH 6.5 for guaiacol [39]. Reproducing the results obtained in the previous validation of the pasteurisation method [18], HTST at 80 °C (CH or RF) did not completely inactivate both oxidase enzymes in salmorejo. PPO activity was initially higher (16–20% RA) than POD activity (11–12% RA), while the decrease in activity over time was somewhat more pronounced for PPO than for POD enzymes. Consequently, salmorejo showed oxidase activities during the first month, and, to a lesser extent, during the second month. Several trials agree that tomato oxidase enzymes cannot be completely inactivated by mild pasteurisation, acting during further storage. Arjmandi et al. (2016) [35] found that POD activity decreased from 30 to 11% RA after 45 days in a pasteurised–chilled smoothie with tomato. Ballesta (2020) [10] also reported decreasing POD activities (from 20 to 8% RA) in pasteurised gazpacho (90 °C for 5 min) kept at 4 °C for 60 days. In another study, a gazpacho treated by heating (50 °C) plus pressurizing (500 MPa for 30 min) showed stable activities for PPO (over 20% RA) than for POD (over 70% RA) during shelf life (30 days at 4 °C) [9]. The oxidase active enzymes that remained in salmorejo after pasteurisation would end up degrading due to their pro-oxidant activity during storage.

Oxidative changes in salmorejo corresponded with the oxidase activities recorded. Browning was detected with the reflectometer (CIELab colour) earlier than with the sensory assessment. The rapid degradation of L-ascorbic acid, a reliable oxidation index used in fruit products [40], confirmed that salmorejo mainly oxidized at the beginning of storage when exposed to oxidase activities responsible for odour/flavour and colour changes. Ballesta (2020) [10] reported that the vitamin C total content (colorimetric determination) decreased from 18 to 13 mg·100 g^−1^ in pasteurised gazpacho (90 °C for 5 min) kept at 4 °C for 60 days. MDA levels increased at the first month, suggesting the greater formation of secondary oxidised lipids (aldehydes, ketones, and others) responsible for rancidity and other flavour alterations. Peroxidation reactions catalysed by POD enzymes might have enhanced lipid secondary oxidation. However, MDA levels were quantitatively low in pasteurised salmorejo, which did not develop rancidity during its shelf life. Fresh odour/flavour predominated over cooked odour/flavour (not included in QDA) as the product was heated for a few seconds at mild temperature. The freshness intensity was low compared to the untreated product (7.1), remained stable during most of storage, and likely decreased due to the accumulation of oxidized volatiles. Salmorejo has a complex flavour chemistry, involving many non-oxidized and oxidized volatiles from tomato, vinegar, garlic, olive oil, and other ingredients generated during processing and further storage. Liu et al. (2022) [12] studied the main aroma-active compounds of fresh and heat-treated (hot break of pulp and sterilisation at 90 °C for 15 min) tomato juice. In this study, different volatiles were associated with fresh (E-2-nonenal), grass (hexanal, 3-hexenal), fruity (6-methyl-5-hepten-one and citral), floral (linalool, phenylacetaldehyde, and b-lonone), and cooked (dimethyl sulphide, dimethyl trisulphide, methional, and 1-octenone) odours. All these and other tomato volatiles can likely be oxidized during the extended shelf life of salmorejo, resulting in detectable flavour changes. The sensory changes resulting from salmorejo oxidation—whether enzymatic or not—took various months to be detected. Enzyme deterioration may have had more impact at the beginning of shelf life, while other oxidative phenomena such as the precipitation of calcium pectates [41], browning caused by Maillard reaction products [42], or the peroxidation of unsaturated lipids probably contributed to the gradual degradation of the product.

HPDE has been proven to be a suitable material for packing salmorejo. The bottle used is translucent and allows the product to be viewed. The appearance of salmorejo, in particular the colour and the separation of emulsion phases, can be examined by consumers before purchasing the product. This is not possible with Tetra Brik or opaque plastic containers. HDPE is stiff, strong, tough, chemical- and moisture-resistant, gas-permeable, and both easy to process and form. HDPE has a high crystallinity level and is less flexible, less transparent, but more resistant to heating than low-density polypropylene predominately used in film applications and in applications where heat sealing is necessary [2]. For the present study, packing plastic material was not required to have certain light and oxygen barrier properties because salmorejo was aerobically processed to obtain a creamy texture and stored in darkness. Moreover, salmorejo was cooled at 25 °C before aseptic filling, avoiding any possibility of the bottle being deformed by heating. It was reported that tomato sauce can be packed at 65–75 °C in HDPE bottles to prevent their deformation without microbiological risks [3]. Whatever the case, the deterioration rate of pasteurised–chilled salmorejo can vary when using plastic materials with different mechanical and thermal properties, or upon different storage conditions (lighting and temperature) that may affect to long-term oxidation reactions.

The shelf life of pasteurised–chilled salmorejo would be limited by a probable loss of its acceptance. Consumer acceptance towards “fresh-like” gazpacho has been studied by Fernández-Ruiz et al. (2017) [11]. This product was the first tomato–oil cold soup industrialised in the 1990s. For the sensory quality of this product, flavour was more decisive than appearance or consistency. Most consumers preferred a gazpacho with a sweet tomato flavour, reminding them of fresh tomato flavour. The use of aromatic vinegar was another positive trait for its acceptability. The conclusions of this study might be extrapolated to salmorejo, a similar tomato–oil homogenate with a less complex flavour as it does not contain fresh pepper, cucumber, or onion. The use of mild preservation treatments is crucial for salmorejo to remain fresh as long as possible. Homemade salmorejo becomes less viscous, and its flavour alters after being refrigerated for 2–3 days. Manufacturers of non-thermally treated salmorejo use extreme aseptic processing conditions and strategies such as acidification or thickening to obtain stable products for a few weeks in refrigeration. Attempts to pasteurise salmorejo with high pressures have provided good sensory results but are relatively effective against enzyme deterioration [6]. The proposed HTST treatments allowed for the retention of freshness attributes and were effective in stabilising sensory quality. The shelf life of salmorejo treated via conventional or radiofrequency heating was similar since pasteurisation is a stage within the common manufacturing process (homogenising, pre-heating, heating, chilling, and packing), and holding times were adjusted in each type of pasteuriser to standardize the thermal effects [18].

The changes which occurred in salmorejo during its shelf life can be summarised as follows (Figure 4). The most intense oxidation mainly corresponds to the first month of storage, likely enhanced by oxidase activities. This initial oxidation is associated with some adverse effects (loss of vitamin C and lipid oxidation) but has no sensory implications. The product remains stable at the physicochemical and sensory levels for the following 3 months, although colour and viscosity changes can be measured with instruments. Between the fourth and fifth month of storage, salmorejo shows clear signs of overall deterioration, including changes in appearance (slight browning and loss of smooth surface), odour/flavour (loss of freshness and homogenisation), and consistency (thinning trend).

## 5. Conclusions

The pasteurised–chilled salmorejo packed in HDPE bottles reached a long shelf life (4 months) compared to that of pasteurised tomato juices or purees. HTST pasteurisation (80 °C for around 10 s) by either conventional or radiofrequency heating can be used to obtain “fresh-like” salmorejo. Product stability is similar when operating conditions are adjusted in both pasteurizers to obtain similar levels of microbiological inactivation. Mesophilic and pathogenic bacteria are easily inhibited by the thermal treatment and when they further undergo chill storage in this acidic product. However, pasteurisation does not prevent salmorejo from showing some enzyme oxidation by PPO and POD at the beginning of its shelf life. Less intense thermal treatments than those referenced until now are required to obtain a stable salmorejo of good sensory quality. The shelf life of pasteurised–chilled salmorejo would be limited by long-term oxidative deterioration and their sensory implications. Future studies will elucidate how storage time affects consumer acceptance of this product. Economic costs and the energy efficiency of radiofrequency technology should be evaluated for its possible implementation in the food industry.

## Figures and Tables

**Figure 1 foods-12-03882-f001:**
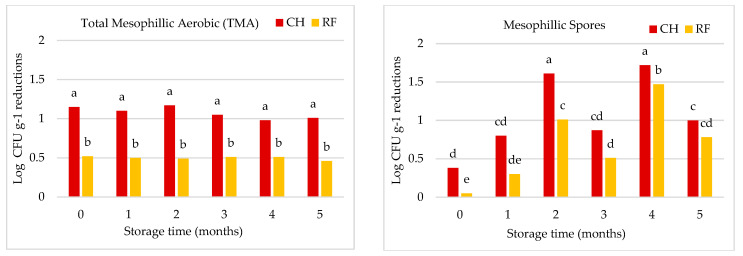
Log·CFU g^−1^ reductions in total mesophilic aerobic microorganisms and spores during the shelf life of salmorejo pasteurised via conventional (CH) or radiofrequency (RF) heating. TMA and spores accounted in raw salmorejo were 4.15 and 2.51 Log·CFU g^−1^ (CH) and 3.90 and 2.98 Log·CFU g^−1^ (RF).^. a–e^ Heating method and storage time effects for *p* < 0.05 (Tukey test).

**Figure 2 foods-12-03882-f002:**
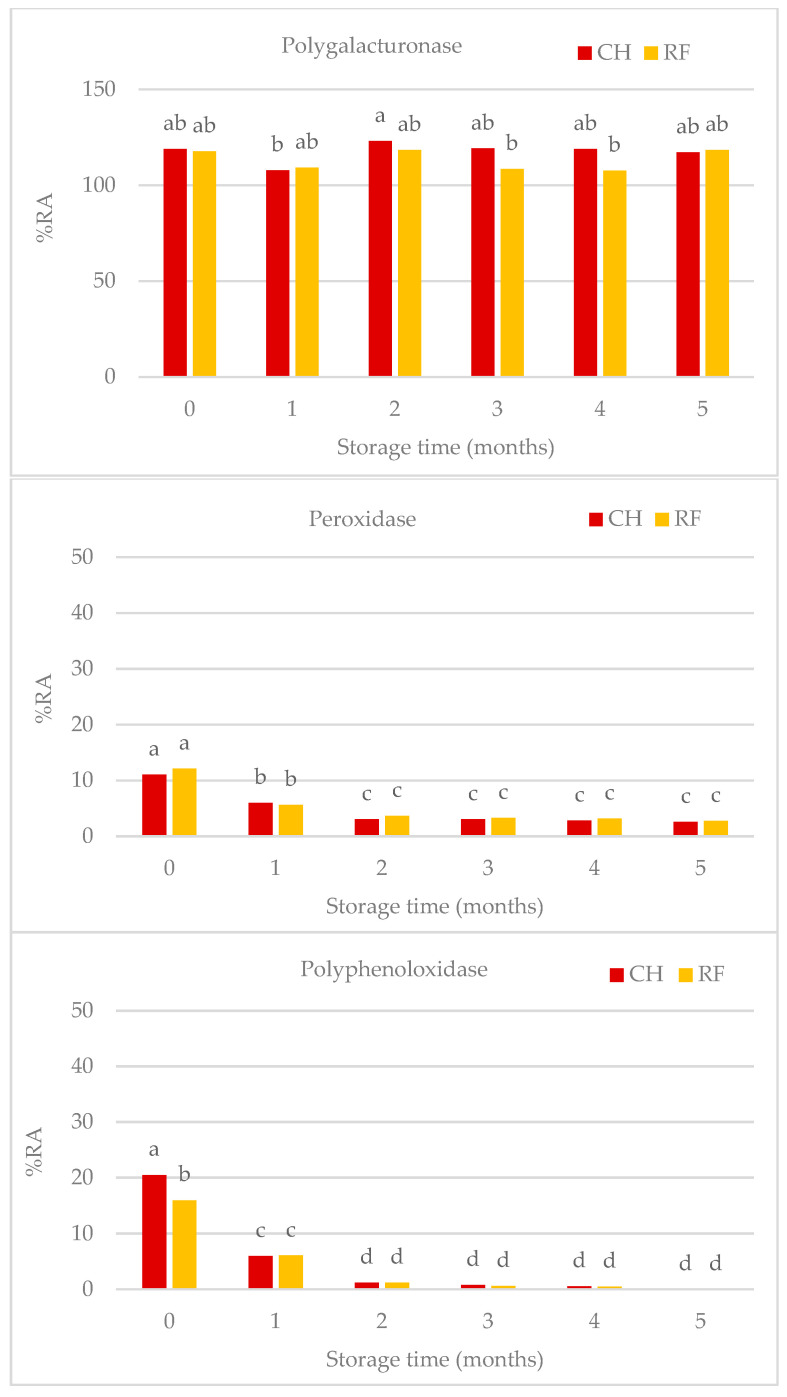
Relative activities (% RA respect to the untreated) of tomato enzymes during shelf life of salmorejo pasteurised via conventional (CH) or radiofrequency (RF) heating. No Pectinmethylesterase activity was found. ^a–d^ Heating method and storage time effects for *p* < 0.05; (Tukey test).

**Figure 3 foods-12-03882-f003:**
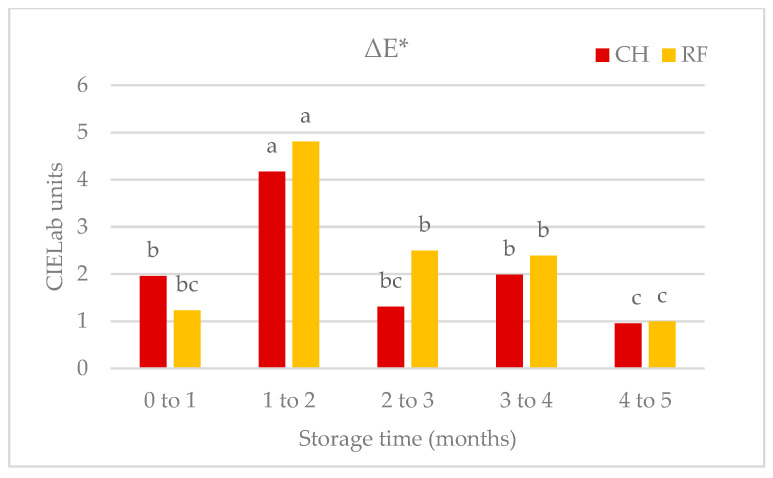
Changes in CIELab colour (ΔE*) of tomato enzymes during the shelf life of salmorejo pasteurised via conventional (CH) or radiofrequency (RF) heating. ^a–c^ Heating method and storage time effects for *p* < 0.05 (Tukey test).

**Figure 4 foods-12-03882-f004:**
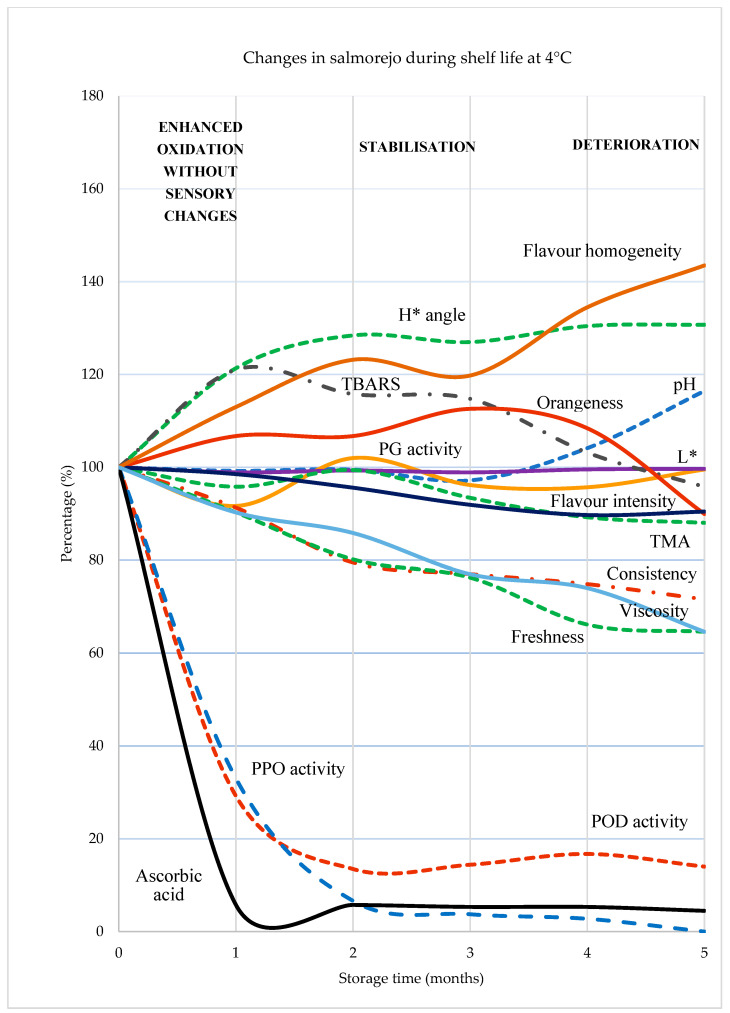
Relative changes (considering the initial values as 100%) of salmorejo quality traits during shelf life. Abbreviations: H*: Hue; TBARS: thiobarbituric reagent substances; PG: polygalacturonase; L*: lightness; TMA: total mesophilic aerobic; PPO: polyphenoloxidase; POD: peroxidase.

**Table 1 foods-12-03882-t001:** Physicochemical changes related to oxidation during shelf life of salmorejo pasteurised via conventional (CH) or radiofrequency (RF) heating.

	Months	0		1		2		3		4		5		
		M		M		M		M		M		M		RMSE
pH	CH	3.97	^bc^	3.88	^bc^	3.82	^bc^	3.78	^c^	4.08	^b^	4.53	^a^	0.333
	RF	3.81	^bc^	3.84	^bc^	3.92	^bc^	3.78	^c^	4.02	^b^	4.53	^a^	
Ascorbic acid	CH	19.7	^a^	1.23	^b^	1.19	^b^	1.15	^b^	1.08	^b^	0.94	^b^	0.086
mg 100 mL^−1^	RF	21.5	^a^	1.12	^b^	1.18	^b^	1.03	^b^	1.10	^b^	0.90	^b^	
TBARS	CH	0.46	^b^	0.58	^a^	0.55	^a^	0.55	^a^	0.48	^b^	0.45	^b^	0.013
mg MDA kg^−1^	RF	0.49	^b^	0.57	^a^	0.55	^a^	0.54	^a^	0.50	^b^	0.46	^b^	
Lightness	CH	60.6	^a^	58.8	^b^	59.6	^b^	59.4	^b^	59.7	^b^	59.7	^b^	0.691
CIE L*	RF	59.2	^b^	58.7	^b^	58.4	^b^	58.1	^b^	58.6	^b^	58.6	^b^	
Redness	CH	22.2	^a^	22.0	^a^	20.3	^b^	20.5	^b^	19.5	^bc^	19.0	^bc^	0.17
CIE a*	RF	21.9	^a^	21.8	^a^	20.0	^b^	20.6	^b^	19.7	^bc^	18.8	^c^	
Yellowness	CH	41.9	^bc^	42.8	^bc^	46.5	^a^	45.3	^ab^	47.0	^a^	46.1	^a^	0.56
CIE b*	RF	39.2	^c^	40.4	^c^	44.8	^ab^	42.4	^b^	44.6	^ab^	45.1	^ab^	
Hue angle	CH	51.7	^b^	62.8	^ab^	66.4	^b^	65.7	^b^	67.5	^b^	67.6	^b^	2.57
CIE h*	RF	52.5	^b^	61.6	^ab^	66.0	^b^	64.1	^b^	66.1	^b^	67.3	^b^	
Viscosity 1	CH	25.0	^a^	20.4	^b^	17.5	^b^	17.9	^b^	18.3	^b^	18.1	^b^	1.08
mPa s	RF	21.7	^ab^	20.3	^b^	17.8	^b^	18.1	^b^	18.8	^b^	17.5	^b^	
Viscosity 2	CH	0.81	^a^	0.78	^a^	0.68	^ab^	0.63	^ab^	0.57	^b^	0.54	^c^	0.135
Pa s	RF	0.82	^a^	0.80	^a^	0.60	^b^	0.58	^b^	0.59	^b^	0.56	^bc^	
Deformation	CH	5.15	^a^	4.93	^ab^	5.35	^a^	5.15	^a^	4.90	^b^	4.90	^b^	0.10
Energy mJ	RF	5.03	^ab^	4.90	^b^	4.70	^b^	5.00	^ab^	5.00	^ab^	5.00	^ab^	
Hardness	CH	0.22	^ab^	0.21	^ab^	0.23	^a^	0.22	^ab^	0.21	^ab^	0.21	^ab^	0.005
N	RF	0.21	^ab^	0.20	^b^	0.20	^b^	0.21	^ab^	0.20	^b^	0.21	^ab^	

Abbreviations: M: mean; RMSE: root mean standard error; TBARS: thiobarbituric acid reactive substances. ^a–c^ Heating method and storage time effects for *p* < 0.05 (Tukey test).

**Table 2 foods-12-03882-t002:** Sensory changes during the shelf life of salmorejo pasteurised via conventional (CH) or radiofrequency (RF) heating.

	Months	0		1		2		3		4		5		
		M		M		M		M		M		M		RMSE
Orange colour	CH	5.9	^a^	6.3	^a^	6.5	^a^	6.5	^a^	6.0	^a^	5.3	^b^	0.20
	RF	6.0	^a^	6.4	^a^	6.2	^a^	6.9	^a^	6.9	^a^	5.4	^b^	
Smooth surface	CH	6.9	^a^	6.8	^a^	6.9	^a^	7.0	^a^	6.3	^a^	5.8	^b^	0.10
	RF	6.5	^a^	6.0	^ab^	6.4	^a^	6.0	^ab^	5.9	^ab^	5.3	^b^	
Odour intensity	CH	5.5		5.5		5.2		4.8		4.8		4.7		0.06
	RF	5.7		5.7		5.9		5.3		5.1		5.2		
Flavour intensity	CH	5.8		5.6		5.5		5.6		5.7		4.7		0.06
	RF	6.0		5.8		6.2		5.1		5.4		4.9		
Odour homogeneity	CH	5.7		5.8		6.1		5.8		5.4		5.2		0.24
	RF	5.8		6.0		6.0		6.0		5.6		5.4		
Flavour homogeneity	CH	5.8		5.6		5.5		5.6		5.7		4.7		0.25
	RF	6.0		5.8		5.5		5.1		5.4		4.9		
Fresh odour	CH	4.1	^a^	3.7	^a^	4.1	^a^	3.9	^a^	3.6	^a^	2.1	^b^	0.17
	RF	4.6	^a^	4.5	^a^	4.5	^a^	4.5	^a^	4.4	^a^	3.4	^b^	
Fresh flavour	CH	4.3	^a^	4.2	^a^	4.2	^a^	4.4	^a^	3.8	^a^	1.8	^b^	0.17
	RF	4.6	^a^	4.3	^a^	4.3	^a^	4.4	^a^	3.1	^a^	2.0	^b^	
Tomato odour	CH	4.0	^a^	3.8	^a^	3.5	^a^	3.6	^a^	3.1	^ab^	2.7	^b^	0.33
	RF	4.8	^a^	4.5	^a^	4.0	^a^	4.1	^a^	3.9	^a^	2.8	^b^	
Vinegar odour	CH	4.2	^a^	4.0	^a^	3.9	^a^	3.7	^a^	3.4	^ab^	2.7	^b^	0.85
	RF	4.4	^a^	4.1	^a^	4.0	^a^	3.8	^a^	3.3	^ab^	3.2	^b^	
Garlic odour	CH	3.4	^a^	3.3	^a^	3.0	^a^	2.9	^a^	2.6	^ab^	2.1	^b^	0.78
	RF	3.4	^a^	3.2	^a^	3.1	^a^	2.8	^a^	2.6	^ab^	2.0	^b^	
Acid taste	CH	4.5	^a^	4.3	^a^	4.3	^a^	4.3	^a^	3.5	^ab^	3.2	^b^	0.04
	RF	4.8	^a^	4.6	^a^	4.9	^a^	4.5	^a^	3.8	^ab^	3.4	^b^	
Mouth feeling	CH	3.7	^a^	3.3	^ab^	3.3	^ab^	3.5	^ab^	3.7	^ab^	2.5	^b^	0.13
	RF	4.3	^a^	3.5	^ab^	3.4	^ab^	3.9	^ab^	3.8	^ab^	2.8	^b^	
Spoon viscosity	CH	5.8	^a^	5.8	^a^	5.2	^ab^	4.8	^ab^	4.9	^ab^	4.1	^b^	0.58
	RF	5.5	^a^	4.2	^ab^	4.5	^ab^	3.9	^b^	4.1	^b^	4.2	^b^	

Abbreviations: M: mean; RMSE: root mean standard Error. ^a,b^ Heating method and storage time effects for *p* < 0.05 (Tukey test).

## Data Availability

The datasets generated and/or analysed during the current study are available from the corresponding author upon reasonable request.

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
