# Peer review of "Factors Limiting Shelf Life of a Tomato–Oil Homogenate (Salmorejo) Pasteurised via Conventional and Radiofrequency Continuous Heating and Packed in Polyethylene Bottles"

_foods, 2023, doi:10.3390/foods12203882_

Round 1

Reviewer 1 Report

The aim of the manuscript is to study the effect of two different heating treatments, conventional heating and radiofrequency, for the pasteurization of salmorejo, and analyze their effect on microbiological and enzymatic activity, physical chemical analysis and sensory analysis during a 5-month period at refrigeration.
In a general way, the article has addressed this objective by performing many different analyses that could be of interest for this specific sample. Furthermore, the presentation of the results is adequate, and the discussion is extense and well explained and supported by literature.
Therefore, I suggest a minor revision of some aspects of the manuscript.
Specific comments of the manuscript are:

Materials and methods
-
Sample preparation
It is mentioned that breadcrumbs were partially substituted by inulin. Has the effect of this component been previously studied over the shelf life of the product? Do the authors think that it will have any specific effect in the product? Specially on the sensory aspect during storage?
-
Conventional heating and radiofrequency pasteurization
Conditions for both treatments are fixed at 80 °C and specific conditions of time and power. How were these conditions selected by the authors? Was there a previous optimization? It is mentioned that conditions were optimized for 5 log pathogen reduction, but not further explanation is provided. Could milder conditions still produce microbiologically safe salmorejo and not affect the product quality? The selection of these treatment conditions should be further explained.
-
Statistical analysis
In this section, it is explained that a factorial design was used to determine the effect of treatments on dependent variables, but not further explanations regarding the results of the experimental design are included along the manuscript.

Results
Regarding the microbiological reduction, shown in Figure 1, apart from showing the log reduction, it would help to follow the changes if also the current load is plotted during each time of measurement, to easily follow the real microbiological load during the storage.
In general, regarding all the analyses results, the data is included for the samples after treatment (0 months) and then for up to 5 months. But the initial data for all the analyses is not included for the untreated salmorejo. Although it is not the aim of this study, the values for the initial prepared salmorejo should be included in all sections. In this way, it will facilitate the understanding of the effect of the two different treatments over the salmorejo, apart from the already reported effect during the shelf life. This data would be easy to compare if included in the tables and figures reporting the different measurements.
As a suggestion for a future work, shorter periods of time between analysis may provide with further information of the trend of the different aspects that were analyzed, especially in the
first month when most of the changes are happening, and towards the end to better determine the shelf life of the product.

Discussion
The authors have performed a good discussion of the results, including many literature references for each of the analysis.

Author Response

Reviewer 1

The aim of the manuscript is to study the effect of two different heating treatments, conventional heating and radiofrequency, for the pasteurization of salmorejo, and analyze their effect on microbiological and enzymatic activity, physical chemical analysis and sensory analysis during a 5-month period at refrigeration.

In a general way, the article has addressed this objective by performing many different analyses that could be of interest for this specific sample. Furthermore, the presentation of the results is adequate, and the discussion is extense and well explained and supported by literature.

Therefore, I suggest a minor revision of some aspects of the manuscript.

Specific comments of the manuscript are:

Materials and methods

Sample preparation

It is mentioned that breadcrumbs were partially substituted by inulin. Has the effect of this component been previously studied over the shelf life of the product? Do the authors think that it will have any specific effect in the product? Specially on the sensory aspect during storage?

Author response (AR). Replacement of bread with inulin had a nutritional objective (less-caloric recipe). In the preliminary trials, inulin addition at 2% did not alter the flavour or texture in freshly made raw or pasteurised salmorejo. Therefore, we did not expect that adding inulin may influence shelf life (which had not been studied). Inulin is not substrate for tomato PME or PG (thinning trend).

Conventional heating and radiofrequency pasteurization

Conditions for both treatments are fixed at 80 °C and specific conditions of time and power. How were these conditions selected by the authors? Was there a previous optimization? It is mentioned that conditions were optimized for 5 log pathogen reduction, but not further explanation is provided. Could milder conditions still produce microbiologically safe salmorejo and not affect the product quality? The selection of these treatment conditions should be further explained.

AR The selection of these conditions was detailed at “Validation of Pasteurisation Temperatures for a Tomato–Oil Homogenate (salmorejo) Processed by Radiofrequency or Conventional Continuous Heating” In Foods 2023, 12(15), 2837.

AR Pasteurization temperatures of 70 to 100 ºC were tested at intervals of 5 degrees using both CH and RF equipment. The flow operating conditions conducted in both equipment (whose pipeline was different) were established on microbiological criteria. Pasteurisation temperature of 80ºC was selected to obtain an adequate enzyme inactivation level (PPO and POD) without product overheating. Using milder conditions also provided microbiologically safe salmorejo but enhanced enzyme deterioration.

AR As described in this paper the thermal treatment was targeted to inactivate Escherichia coli O157:H7, Salmonella enterica and Listeria monocytogenes to ensure at least a 5-Log reduction of vegetative pathogens of concern. Due to low pH of the product, below 4.5, milder conditions might have been applied.  According to US Food and Drug Administration (FDA) specifications, a minimum temperature–time of 71.1 °C for 3 s is sufficient to achieve a 5-Log reduction of vegetative bacterial pathogens in acidic products (pH < 4) (FDA, 2004) and the HTST conditions used in the present study exceed these specifications.

AR Specific information on the pasteurisation methods has been now included. See line 108. For more information, please, see our article.

AR Dufort, E.L.; Etzel, M.R.; Ingham, B.H. Thermal Processing Parameters to Ensure a 5-Log Reduction of Escherichia coli O157:H7, Salmonella Enterica, and Listeria monocytogenes in Acidified Tomato-Based Foods. Food Prot. Trends 2017, 37, 409–418

AR FDA Guidance for Industry: Juice Hazard Analysis Critical Control Point Hazards and Controls Guidance, First Edition; Department of Health and Human Services, Food and Drug Administration: Washinton, DC, USA, 2004.

Statistical analysis

In this section, it is explained that a factorial design was used to determine the effect of treatments on dependent variables, but not further explanations regarding the results of the experimental design are included along the manuscript.

AR Statistical model is not correctly indicated. We did an ANOVA with a repeated measures design as corresponds to samples from the same manufacturing batch maintained over time. In addition, group means (RF vs CH at the same time) were checked using one-way ANOVA to confirm the effects. In fact, practically, result for CH and RF were similar at the same storage time. See line 246.

Results

Regarding the microbiological reduction, shown in Figure 1, apart from showing the log reduction, it would help to follow the changes if also the current load is plotted during each time of measurement, to easily follow the real microbiological load during the storage.

AR Using Log CFU/g reductions is a standardised procedure. According to your request, we have included the microbiological counts of raw salmorejo (CH and RF) in figure 1 footnote. See line 272. We expect that results can be better followed.

In general, regarding all the analyses results, the data is included for the samples after treatment (0 months) and then for up to 5 months. But the initial data for all the analyses is not included for the untreated salmorejo. Although it is not the aim of this study, the values for the initial prepared salmorejo should be included in all sections. In this way, it will facilitate the understanding of the effect of the two different treatments over the salmorejo, apart from the already reported effect during the shelf life. This data would be easy to compare if included in the tables and figures reporting the different measurements.

AR Our objective was to study the pasteurised-chilled product. For this reason, changes in microbial loads and in enzyme activities were expressed with respect to raw product. Moreover, viscosity could not be measured upon the same conditions in raw and heated samples. No data on raw product are available for these variables. Figure 4 provides a summarised explanation based on the relative percentages with respect to the freshly pasteurized product (day 0). For example, heating resulted in changes of colour (orangeness), flavour (loss of freshness) and consistency (thickening)m while chill storage resulted in browning, flavour alteration and thinning. We think that introducing data on raw product may create some confusion.

AR Pasteurization effects were evaluated at “Validation of Pasteurisation Temperatures for a Tomato–Oil Homogenate (salmorejo) Processed by Radiofrequency or Conventional Continuous Heating” In Foods 2023, 12(15), 2837.

As a suggestion for a future work, shorter periods of time between analysis may provide with further information of the trend of the different aspects that were analyzed, especially in the first month when most of the changes are happening, and towards the end to better determine the shelf life of the product.

AR Thanks for your comment. This trial was planned with monthly controls because salmorejo can have a shelf life of up to six months. Indeed, we agree that control times should be lower in future shelf-life studies.

Thank you very much for your comments.

Reviewer 2 Report

This manuscript describes two methods of insuring the microbiological safety of salmorejo along with how the product changes with storage. It is an interesting study that should bring some attention. 

The abstract should emphasize the differences observed between products that were pasteurized by conventional and radiofrequency heating. This is the primary idea behind the study.

Line 77: L. monocytogenes, with a lower case m.

Line 82: A period is missing after the sentence.

Lines 115 and 119 and later: SI unit abbreviations are mL and mo.

Line 187: butylatedhydroxytoluene

Line 346: range from 3.9-4.4

Line 357: confirmed instead of checked

Line 422: corresponded with instead of were coherent with

Line 424: L-ascorbic

Figure 4 and its explanation belong at the end of Discussion and not in Conclusions

The language is good, but could be improved in several places. The editor can take care of this.

Author Response

Reviewer 2

This manuscript describes two methods of insuring the microbiological safety of salmorejo along with how the product changes with storage. It is an interesting study that should bring some attention. 

The abstract should emphasize the differences observed between products that were pasteurized by conventional and radiofrequency heating. This is the primary idea behind the study.

Line 77: L. monocytogenes, with a lower case m.

AR (Author response) Mistake amended. See line 77.

Line 82: A period is missing after the sentence.

AR Time period introduced. See line 82.

Lines 115 and 119 and later: SI unit abbreviations are mL and mo.

AR Mistake amended. See throughout the manuscript.

Line 187: butylatedhydroxytoluene

AR Mistake amended. See line 192.

Line 346: range from 3.9-4.4

AR Mistake amended. See line 351.

Line 357: confirmed instead of checked

AR Changed. See line 362.

Line 422: corresponded with instead of were coherent with

AR Changed. See line 427.

Line 424: L-ascorbic

AR Changed. See line 429.

Figure 4 and its explanation belong at the end of Discussion and not in Conclusions

AR Related text has been moved to the end of discussion. See line 494.

Thank you very much for your comments.

Reviewer 3 Report

Dear Authors, Your work is concise but exhaustive for the great number of determinations carried out, even if no particular attention is focused on emerging technologies. 

I have some doubts concerning the plastic bottle used, typically used  for biological analyses and not for food drink and fluid food products. HDPE is not so transparent to allow a perfect sight of the product, such as PET bottle (which are a standard for water, fruit juice and so on). Anyway, the aim of Your work was not to demonstrate the influence of the packaging material on the quality evolution of salmorejo, so it is ok. Concerning Your conclusions, in my opinion You have to emphasize the difference among traditional heat treatments with the proposed ones.

Author Response

Reviewer 3

Dear Authors, Your work is concise but exhaustive for the great number of determinations carried out, even if no particular attention is focused on emerging technologies.

I have some doubts concerning the plastic bottle used, typically used  for biological analyses and not for food drink and fluid food products. HDPE is not so transparent to allow a perfect sight of the product, such as PET bottle (which are a standard for water, fruit juice and so on). Anyway, the aim of Your work was not to demonstrate the influence of the packaging material on the quality evolution of salmorejo, so it is ok. Concerning Your conclusions, in my opinion You have to emphasize the difference among traditional heat treatments with the proposed ones.

Author response (AR) As mentioned, this was not our objective, because testing different packs would had implied introducing an additional treatment. The most common packs used in salmorejo are bricks and PE bottles. We chose HDPE bottles with intermediate translucency and thermal resistance. Oxidation rate of salmorejo may vary depending on the pack, temperature or lighting, but it is clear that long-term oxidation is the main limiting factor of its shelf life.

AR Main difference among the referenced and the proposed heat treatments are now explained in the conclusion section. See lines 507-517.

Thank you very much for your comments.

Reviewer 4 Report

After reviewing the original article on the shelf life of salmorejo subjected to various pasteurization methods, I find the article to be of good quality. The organization is commendable, the details are thoroughly described, and the scientific sound is of good quality. The results are well-discussed, providing valuable insights into the factors affecting the shelf life of salmorejo. However, I would recommend the inclusion of a diagram illustrating the experimental protocol for better clarity. 

Overall, the article is a valuable contribution to the field.

A thorough check for minor grammatical errors would further enhance the manuscript's quality. 

Author Response

Reviewer 4

After reviewing the original article on the shelf life of salmorejo subjected to various pasteurization methods, I find the article to be of good quality. The organization is commendable, the details are thoroughly described, and the scientific sound is of good quality. The results are well-discussed, providing valuable insights into the factors affecting the shelf life of salmorejo. However, I would recommend the inclusion of a diagram illustrating the experimental protocol for better clarity. 

Overall, the article is a valuable contribution to the field.

Author respose (AR) A flow diagram of manufacturing process was included in the graphical abstract. Original diagram with photos appears at “Validation of Pasteurisation Temperatures for a Tomato–Oil Homogenate (salmorejo) Processed by Radiofrequency or Conventional Continuous Heating” In Foods 2023, 12(15), 2837.

Thank you very much for your comments.